# Lysosomal Membrane Stability of Mussel (*Mytilus galloprovincialis* Lamarck, 1819) as a Biomarker of Cellular Stress for Environmental Contamination

**DOI:** 10.3390/toxics11080649

**Published:** 2023-07-26

**Authors:** Elena-Daniela Pantea, Valentina Coatu, Nicoleta-Alexandra Damir, Andra Oros, Luminita Lazar, Natalia Rosoiu

**Affiliations:** 1Ecology and Marine Biology Department, National Institute for Marine Research and Development “Grigore Antipa”, 300 Mamaia Blvd., 900581 Constanta, Romania; 2Institute of Doctoral Studies, Doctoral School of Applied Sciences, Biology Domain, Ovidius University of Constanta, 58 Ion Vodă Street, 900573 Constanta, Romania; 3Chemical Oceanography and Marine Pollution Department, National Institute for Marine Research and Development “Grigore Antipa”, 300 Mamaia Blvd., 900581 Constanta, Romania; vcoatu@alpha.rmri.ro (V.C.); ndamir@alpha.rmri.ro (N.-A.D.); aoros@alpha.rmri.ro (A.O.); 4Preclinical Disciplines Department, Biochemistry Domain, Faculty of Medicine, Ovidius University of Constanta, Campus B, 1 University Alley, 900470 Constanta, Romania; natalia_rosoiu@yahoo.com; 5Biological Sciences Section, Academy of Romanian Scientists, 3 Ilfov Street, Sector 5, 50044 Bucharest, Romania

**Keywords:** *Mytilus galloprovincialis*, lysosomal stability, neutral red retention time, hemocytes, chemical pollution, Black Sea

## Abstract

The lysosomal membrane stability (LMS) of hemocytes in wild mussels (*Mytilus galloprovincialis*) as a biomarker of cellular stress for chemical pollution was tested by neutral red retention time (NRRT) assays. To assess the environmental contamination in the study area, seawater quality and pollutant bioaccumulation throughout the soft tissue of mussels were investigated. The samples were collected in July 2022 at four sites on the Romanian Black Sea coast considered to be differently affected by contamination. To support the suitability of LMS as a biomarker of contaminant-induced stress, the contaminant body burdens of the mussels were compared with the NRRT values. The results showed a significantly reduced NRRT in all investigated locations, particularly in port areas (mean retention time between 11 and 14 min). The elevated bioaccumulation of organochlorinated pesticides (OCPs) and polychlorinated biphenyls (PCBs) and low NRRTs were observed at the most contaminated sites (i.e., ports). The low lysosomal stability reflected stress and damage in the hemocytes of mussels and could be related to the body burdens of contaminants. LMS is an effective indicator of health status in mussels and could be considered a sensitive biomarker of cellular stress induced by contaminant exposure.

## 1. Introduction

The marine environment has been subjected to increasing anthropogenic activities that have generated a large variety of pollutants. In the past, the chemical contamination of coastal ecosystems has been addressed only in terms of the type and level of the most common pollutants such as heavy metals, hydrocarbons, and organochlorinated compounds, but such analyses have provided no information on their harmful effects [1,2].

Mussels, filter-feeding and sessile bivalve molluscs, are useful sentinel organisms to assess the chemical pollution of the surrounding waters because of their ability to accumulate different classes of pollutants, thus providing a time-integrated image of their bioavailability [3,4,5]. To establish the status of marine environmental and organism health, several sensitive biomarkers (i.e., lysosomal membrane stability, lysosomal lipofuscin content, lysosomal neutral lipids, peroxisomes proliferation, total oxidant scavenging capacity, acetylcholinesterase activity, and metallothionein content) have been developed and applied to assess stress syndromes caused by chronic exposure to pollutants in an aquatic environment [1,6,7,8,9,10]. 

Biomarkers that reflect a rapid contaminant-induced response at the molecular, biochemical, and cellular levels have often been considered sensitive ‘early warning’ indicators of organisms’ responses to toxic chemicals [1,11]. Contaminants induce sublethal cellular pathologies that cause the perturbation of function and structure at the molecular level [12,13,14,15]. The early signs of stress are associated with changes in subcellular organelles such as lysosomes, the endoplasmic reticulum, and mitochondria [12] and could serve to anticipate damage at the population level [6,16].

Lysosomes are widely present in the digestive cells and hemocytes of mussels and play a key role in cell physiology, food digestion, immunity function, autophagic cellular turnover, sequestration, and the excretion of harmful compounds [13]. Lysosomes have been acknowledged as a specific target for the toxic effect of many pollutants because of their high capacity for pollutant accumulation [2,11,13,17]. Lysosomes can accumulate heavy metals and organic compounds such as PAHs, PCBs, and OCPs and pharmaceutical substances [11,12,14,15,18]. Lysosomes are extremely sensitive to minimal concentrations of contaminants and show an exceptionally rapid response [5,17].

Lysosomal membrane stability (LMS) as a general stress biomarker of chemical pollution has been widely recommended by different organizations such as the Barcelona Convention, International Council for Exploration of the Sea (ICES), and Oslo and Paris Conventions (OSPAR) [17]. 

Only a few field studies have tested the LMS of wild bivalves on the Romanian Black Sea coast in the last two decades. LMS assays were applied to *Mytilus galloprovincialis* [19,20] and the soft-shelled clam (*Mya arenaria*) [21].

The main aim of this study was to investigate the cellular well-being of *M. galloprovincialis* living hemocytes via the neutral red retention time (NRRT) assay in order to evaluate their health status after chronic exposure to different environmental concentrations of pollutants, and, further, to link the total contaminant (heavy metals, organochlorinated pesticides, polychlorinated biphenyls, and polycyclic aromatic hydrocarbons) body burdens with the lysosomal neutral retention time. In parallel, the water quality and the degree of contaminant bioaccumulation in the soft tissue of the mussels in the study area were assessed.

## 2. Materials and Methods

### 2.1. Study Area and Sampling Procedure

Four field samplings were conducted along the Romanian Black Sea coast in July 2022. The study areas were chosen based on their suitability for assessing the biological effects of exposure to various anthropogenic contaminants associated with urban areas and ports. Considering that this 225 km coastline experiences major impacts from the Danube’s discharge in the northern area and industrialized and urbanized areas, as well as being intensely touristy in the summer in the southern area, we did not identify a pristine area for the implementation of the study.

Seawater and biota (*M. galloprovincialis*) were simultaneously sampled from each site. The location of the sampling sites and the associated pressures are presented in Table 1. The study area map was generated with ArcGIS Desktop 10.7 software [22] (Figure 1).

The seawater temperature was recorded in situ at each sampling site with a calibrated glass technical thermometer (Termodensirom). Seawater samples were collected for subsequent laboratory analyses of the abiotic factors (salinity, pH, dissolved oxygen) and contaminants such as total petroleum hydrocarbons (TPHs), polycyclic aromatic hydrocarbons (PAHs), organochlorinated pesticides (OCPs), polychlorinated biphenyls (PCBs), and heavy metals (HMs).

Mussels (*M. galloprovincialis*) were randomly sampled using a modified rake with a net, from a depth of 0.5 to 1 m, and transported to the laboratory in a cool box with ice packs and ambient seawater from the sampling site within 0.5–1.5 h, depending on the distance between the site and laboratory. Fifteen mussels were selected for the LMS assay from each site (60 specimens in total) within the size range of 4–6 cm shell length, to reduce the possible effect of specimen size and sexual maturity on the response of this biomarker. At each sampling site, 20–25 mussels were also taken for chemical analysis. 

### 2.2. Environmental Data

Salinity and pH were measured in the laboratory using a Mettler Toledo^TM^ S479 (SevenExcellence^TM^ Multiparameter Benchtop, ©Mettler-Toledo GmbH, Greifensee, Switzerland). Dissolved oxygen (DO) was determined by the Winkler method (SR EN 25813:2000) according to the “Methods of Seawater Analysis” manual [23]. The method uses iodometric titration and relies on the ability of dissolved oxygen in the sample to gradually oxidize the added reagents. Data quality was ensured by determining the sodium thiosulfate solution factor before each set of analyses. 

### 2.3. Chemical Analyses 

#### 2.3.1. Sample Processing

Soft tissues from 20–25 mussels were dissected and washed with seawater and distilled water. All soft tissues were pooled into a composite sample and freeze-dried (Freeze Dry System, FreeZone 2.5, LABCONCO, Kansas City, MO, USA). After drying, the samples were homogenized using an electric grinder. 

#### 2.3.2. Organic Contaminants in Seawater and Tissue

Sixteen PAHs (naphthalene, acenaphthylene, acenaphthene, fluorene, anthracene, phenanthrene, fluoranthene, pyrene, benzo[a]anthracene, chrysene, benzo[a]pyrene, benzo[b]fluoranthene, benzo[k]fluoranthene, benzo[g,h,i]perylene, indeno [1,2,3-cd], pyrene, and dibenzo[a,h]anthracene); nine OCPs (hexachlorobenzene (HCB), lindane, heptachlor, aldrin, p,p′-DDE, dieldrin, endrin, p,p′-DDD, and p,p′-DDT); and seven PCBs (PCB 28, PCB 52, PCB 101, PCB 118, PCB 153, PCB 138, and PCB 180) were investigated in seawater and mussel tissue.

The analytical method for the determination of organic contaminants in seawater and mussel tissues was carried out according to the protocol in [24]. The extraction of PAHs, OCPs, and PCBs from seawater samples was performed in a separatory funnel with a mixture of hexane/dichloromethane (7/3; *v*/*v*).

For PAH analysis, about 2 g of the freeze-dried tissue was Soxhlet-extracted for 8 h with 250 mL of methanol. The extracts were then saponified by adding 20 mL of 0.7 M KOH and 30 mL of deionized water and refluxed for 2 h. The resulting mixture was transferred into a separating funnel and extracted three times with hexane (once with 90 mL and twice with 50 mL). Then, the extracts were mixed, filtered on glass cotton, and dried on anhydrous sodium sulphate. The extracts were concentrated by rotary evaporation (Laborota 4001) down to a volume of 15 mL. For PAH analysis, seawater and tissue extracts were further processed by clean-up in a silica/alumina column and elution with 20 mL of hexane, 30 mL of a hexane/methylene chloride (90/10; *v*/*v*) mixture, and 20 mL of hexane/methylene chloride (50/50; *v*/*v*). The eluents were combined and concentrated to 15 mL using the Kuderna–Danish concentrator and then reconcentrated under nitrogen flow to about 1 mL using a nitrogen concentrator (TurboVap LV Evaporator, Biotage, Sweden). The analytical determination of the PAH content in seawater and mussel tissue samples was carried out by the gas-chromatographic method with a Perkin Elmer CLARUS 690 Gas Chromatograph, coupled with a mass spectrometer detector (GC-MS). Internal standard 9,10-dihydroanthracene was added to the samples to quantify the analytical procedures’ overall recovery. One extraction blank was performed in each series of analyses. The recoveries of PAHs were between 0.001% and 0.06% for seawater samples, and for mussel samples they were between 0.01% and 0.073%. The detection limit for PAH content was 0.1 ng/L for seawater and 0.1 ng/g dry weight (dw) for tissue samples.

About 2 g of freeze-dried tissues was used for the extraction of organochlorinated compounds (OCPs and PCBs) from mussel tissue. The organochlorinated compound extraction was carried out with 30 mL hexane/acetone (1/1; *v*/*v*) in the microwave extraction system (Milestone START E) for 30 min at 120 °C. For organochlorinated compound analysis, seawater and tissue extracts were further processed following several steps: extract concentration by a rotary evaporator (Laborota 4001), clean-up using florisil, and the concentration of the samples using a Kuderna–Danish concentrator and nitrogen flow to a volume of about 1 mL. The analytical determination of the organochlorinated compounds was carried out by the gas-chromatographic method with a Clarus 500 Gas Chromatograph (PerkinElmer, Waltham, MA, USA) equipped with an electron capture detector (GC-ECD). To quantify the overall recovery of the analytical procedures, internal standard 2,4,5-trichlorobenzene was added to the samples. One extraction blank was performed in each series of analyses. The OCP recoveries from seawater were between 5.47% and 19.21%, and from mussels they were between 200.81% and 288.37%. The PCB recoveries from seawater were between 139.93% and 280.72%, and for mussels they were between 71.72% and 114.79%. The detection limits of organochlorinated compounds were in the range of 2–4 ng/L for OCPs and 3–6 ng/L for PCBs for seawater samples, and 0.2–0.5 ng/g dw for OCPs and 0.3–0.7 ng/g dw for PCBs for mussel samples.

The gas chromatographs were equipped with an Elite 35 MS capillary column ((35%-diphenyl)-dimethylpolysiloxane; 30 m l. × 0.32 mm i.d. × 0.25 mm f.t). The chromatographic conditions were as follows: carrier gas—helium (1 cm^3^/min speed, 15 cm^3^/min split flow); injector temperature—300 °C; oven temperature program—initial temperature 180 °C, ramp 1 (7 °C/min to 230 °C and hold 10 min), ramp 2 (15 °C/min to 250 °C and hold 2 min); detector temperature—330 °C.

Total petroleum hydrocarbons (TPHs) from seawater samples were extracted with a mixture of hexane/dichloromethane (7/3; *v*/*v*) in a separatory funnel and analyzed by the fluorometric method with a FLUORAT 02-3M fluorimeter analyzer.

PAH, OCP, and PCB concentrations in seawater are reported as ng/L and in tissue as µg/kg tissue dry weight (dw).

#### 2.3.3. Heavy Metals in Seawater and Tissue

Seawater concentrations of copper (Cu), cadmium (Cd), lead (Pb), nickel (Ni), and chromium (Cr) were determined from unfiltered and acidified water samples (acidified up to pH = 2 with HNO3 Ultrapure). For heavy metal (Cu, Cd, Pb, Ni, Cr, and Co) analyses in tissue, about 0.5 g of dry tissue was digested with 5 mL concentrated nitric acid in sealed Teflon vessels on an electric hot plate at 120 °C. The solution was made up to 100 mL with deionized water (18.2 MΩ.cm, Millipore, Burlington, MA, USA). Heavy metal (HM) determinations were performed on an atomic absorption spectrometer (HR-CS ContrAA 800 G equipment, Analytik Jena, Jena, Germany). Calibration was performed with working standards prepared from Merck stock solutions for each element in the following ranges: 0−50 µg/L (Cu), 0−10 µg/L (Cd), 0−25 µg/L (Pb), 0−50 µg/L (Ni), 0−50 µg/L (Cr), and 0−25 µg/L (Co). Each sample was measured in three parallel sub-samples, and the average value was reported. The method detection limits for HMs were between 0.001 and 0.01 µg/L in seawater samples and 10 and 100 µg/kg tissue dry weight in mussels. To ensure the accuracy of the analytical procedures, standard protocols were used [23,25]. The seawater concentrations of heavy metals are expressed as µg/L and the tissue concentrations as µg/kg tissue dry weight (dw).

### 2.4. Lysosomal Membrane Stability Assessment

Mussels were kept unfed for about 24 h at 20 ± 1 °C in 5 L plastic vessels containing constantly aerated ambient seawater before extracting the hemolymph. A stock solution of neutral red (NR; 100 mM) was prepared by dissolving 28.8 mg of NR dye powder (N7005-1G, dye content ≥ 90%, SIGMA-ALDRICH, Burlington, MA, USA) in 1 mL of dimethyl sulfoxide (DMSO). A working solution was obtained by diluting 10 μL of the stock solution in 5 mL physiological saline.

LMS was evaluated by the neutral red retention time (NRRT; min) assay in hemocytes of mussels following the in vivo cytochemical method described by Martínez-Gómez et al. [17]. The principle of this test is based on the ability of healthy lysosomes to retain the dye longer than affected ones; lysosomal damage can cause the leakage of the NR dye into the cytosol, possibly leading to cell death [6].

The mussel valves were pried apart with a solid scalpel, and 0.1 mL hemolymph was withdrawn from the posterior adductor muscle with a 1 mL hypodermic syringe fitted with a 21-gauge needle containing 0.1 mL of physiological saline solution (NaCl, 9 mg/mL, BRAUN). After obtaining the hemolymph sample, the needle was discarded to reduce shearing forces during the expulsion of the syringe contents into a 2 mL Eppendorf microtube held in ice water.

A quantity of 50 μL of the hemolymph/physiological saline mixture was dispensed onto a microscope slide previously treated with 2 μL of poly-L-lysine (a coating agent for cell adhesion on slides) and suspended on a rack in a dark humid chamber consisting of a cool box containing a mixture of water and ice on the bottom. The cells were left to settle and attach to the slide surface for 15 min. After the excess solution was carefully drained, 40 μL of NR working solution was added to the area containing the attached cells and a coverslip was applied. After 15 min of incubation in the dark humid chamber, the slides were examined systematically under an OLYMPUS IX73 inverted microscope (×400 magnification) at 15, 30, 60, 90, 120, 150, and 180 min. Following each inspection, the slides were returned to the dark humid chamber. The endpoint of the test was when more than 50% of the hemocytes exhibited dye loss into the cytosol.

At the same time, the cells were examined for lysosomal alterations and were given a score (from 0 to 5) according to the severity of the effect: score 0 = no effects; score 1 = enlargement but no leakage; score 2 = leakage but no enlargement; score 3 = leakage and enlargement; score 4 = leakage and enlargement but colorless lysosomes; score 5 = rounded up fragmenting cells [17].

The percentage of LMS (or lysosomal damage) was calculated using the scoring procedure. The weighted score was consequently calculated by multiplying the score with the weighting factor for that time. The scoring was truncated at 120 min. The total score of the lysosomal condition was estimated as:$$\%\;LMS=\left(1-\left(\frac{\sum \;ws}{75}\right)\right)\times 100$$
where ∑ *ws* is the sum of weighted scores, calculated by multiplying the score by the weighting factor for that time.

NRRT was assessed against the background assessment criteria (BAC) and environmental assessment criteria (EAC) [17]. The threshold values for the NRRT assay were as follows: no stressed organisms or healthy organisms if NRRT ≥ 120 min (BAC); stressed but compensating if 120 min > NRRT ≥ 50 min (BAC); severely stressed and probably exhibiting pathology if <50 min (EAC).

### 2.5. Data Analysis

Biomarker data were tested for normal distribution (Shapiro–Wilk test) and the homogeneity of variances (Levene’s test). Differences in biomarker response (lysosomal NRRT) between sites were tested for statistical significance using a Kruskal–Wallis ANOVA on ranks with Dunn’s post hoc test. The significance level of the statistical results was set at *p* < 0.05. Multivariate statistical analysis (principal component analysis, PCA) was applied for the interpretation of data variability. The principal components (PCs) considered in the analysis were those with eigenvalues greater than one. PCA was conducted to assess the relationship between seawater and tissue contaminant levels and the biomarker values. The sum of PAH, OCP, and PCB compounds and the mean values of the biomarker was considered for the PCA test. Statistical analyses of data were carried out with XLSTAT 2021.2.1 software (New York, NY, USA) [26]. The shade plot, a visual representation of the data matrix, was constructed with PRIMER v7.0.21 (PRIMER-E Ltd., Plymouth, UK) [27]. The different scale intensities in the shaded plot show the data pattern. Due to the large variations, all data were square-root transformed.

## 3. Results

### 3.1. Environmental Conditions in the Study Areas

Information regarding the environmental conditions of each sampling site is presented in Table 2. The seawater temperature showed specific values for the summer season, with values between 23 °C (at MDP) and 26 °C (at MGP). The lowest salinity value was recorded at MPB (13.81 psu) and the highest at MGP (15.08 psu), a normal range for the Black Sea’s brackish water surface. The oxygen level at MPB was slightly below the minimum accepted level (6.2 mg/L) according to the national legislation for the quality of surface waters [28]. High oxygen levels were observed for MDP and MGP. This could have been related to phytoplankton blooms, common events occurring in port areas due to favorable conditions. The pH values varied between 8.07 (in MPB) and 8.63 (in MGP).

### 3.2. Organic Contaminant Concentration in Seawater and Tissue

The seawater concentrations of organic pollutants are presented in Figure 2 and Appendix A. The TPH levels varied between 2 µg/L (in MPB) and 1235,42 µg/L (at MDP). The highest seawater concentrations of ∑PAH (6.50 µg/L) and ∑PCB (0.20 µg/L) were detected at CTP and of ∑OCP (98.79 µg/L) at MGP. Six compounds were detected among the sixteen PAHs investigated in the seawater samples: naphthalene, fluorene, phenanthrene, anthracene, fluoranthene, and pyrene. The values of the dominant compounds (naphthalene, phenanthrene, and anthracene) varied between the detection limit and 4.37 µg/L, the highest value being measured for phenanthrene at CTP. We recorded the highest values for p,p′ DDD, in particular at MGP (76.13 µg/L), MPB (19.85 µg/L), and MDP (17.69 µg/L). Also, elevated values of p,p′-DDE, p,p′-DDT, heptachlor, dieldrin, and endrin were measured at MPB. Among the investigated PCBs in seawater, only PCB 52, PCB 101, and PCB 118 showed values that exceeded the detection limit. The rest of the organic compounds recorded concentrations near or below the detection limit.

The tissue concentrations of organic pollutants are presented in Figure 2 and Appendix A. The concentrations of ∑PAH in mussels were low and ranged between 1.57 and 2.26 µg/kg dw; the maximum value being recorded at MPB. The ∑OCP and ∑PCB bioaccumulation in mussels was exceptionally high at all sampling sites. The highest level of ∑OCP (3952.39 µg/kg dw) and ∑PCB (8450.46 µg/kg dw) was measured at MDP. Phenanthrene and anthracene, low-molecular-weight (LMW) PAHs, showed the most elevated values. Increased values were detected for heptachlor (at all sampling sites) and lindane (at MDP). PCB 52 recorded exceptionally high values at all sampling sites. High concentrations were also determined for PCB 118, PCB 101, and PCB 180. The other chemical compounds recorded levels near or below the detection limit.

### 3.3. Heavy Metal Concentration in Seawater and Tissue

The seawater concentrations of heavy metals greatly fluctuated between sites (Figure 2, Appendix A). The highest concentrations of Cu (14.710 µg/L) were detected at MPB; of Cd (0.017 µg/L) and Pb (2.757 µg/L) at MGP; of Ni (1.120 µg/L) at MDP; and of Cr (15.780 µg/L) at CTP. Cu, Pb, and Ni were the most bioaccumulated metals in tissue (Figure 2, Appendix A). The highest concentrations of Cu (6599 µg/kg dw), Cd (1331 µg/kg dw), Pb (2755 µg/kg dw), Ni (2712 µg/kg dw), and Co (305 µg/kg dw) were detected at MPB. In contrast, Cr accumulated more in the soft tissue of mussels from CTP (1341 µg/kg dw) (Figure 2, Appendix A). Increased heavy metal (∑HM) concentrations in seawater and tissue were measured at MPB.

### 3.4. Lysosomal Membrane Stability

The average, minimum, and maximum values of the NRRT of lysosomes are illustrated in Figure 3. The highest mean (mean ± standard deviation) NRRT was detected at MPB (34 ± 18.34 min). Mussels from CTP (11 ± 10.56 min), MGP (12 ± 10.14 min), and MDP (14 ± 10.56 min) showed an extremely reduced capacity to retain the dye.

The NRRT ranged between 0 and 60 min in all samples, showing the low LMS of the analyzed mussels (Figure 4). The NRRT in the lysosomes of mussels were as follows: 15 min for 25 specimens, 30 min for 15 specimens and 60 min for 4 specimens. No NR retention (0 min) was recorded in 16 individuals who showed severe lysosomal membrane destabilization, confirmed by the immediate loss of NR dye in their cytosol. Mussels collected from port areas revealed a faster lysosomal neutral loss (more than 47% of individuals showed an intralysosomal NR loss during the first 15 min of the assay) than the mussels from MPB.

The highest mean lysosomal stability of mussels (49.16 ± 11.13%) was seen at MPB and the lowest (30.67 ± 6.96%) at MDP. In general, a high degree of lysosomal damage was observed in all analyzed specimens (Figure 5).

The Shapiro–Wilk test showed that the retention time data significantly deviated from a normal distribution (*W*_(59)_ = 0.823, *p* = 0.0001). The homogeneity of variances (Levene’s test) showed that there was no difference between the variances (*F*_(3, 56)_ = 2.76, *p* = 0.108). Based on this outcome, data were tested for site differences using a non-parametric test (Kruskal–Wallis ANOVA on ranks). The test results showed significant differences between the NRRTs of mussels at all sampling sites (*p* = 0.0003). Further, the results were tested for site-specific differences by multiple pairwise comparisons (Dunn’s post hoc test). Statistically significant differences were detected between MPB and port sites: MDP (*p* = 0.002), CTP (*p* = 0.001), and MGP (*p* = 0.001) (Appendix A).

The PCA of the mean concentration of contaminants and abiotic factors in seawater, the mean NRRT, and the related variability of the site distribution are shown in Figure 6. Eigenvalues of 7.73 and 4.92 were found for the first two principal components. Principal component 1 (PC1) and principal component 2 (PC2) of the PCA explained 90.43% of the total variance in the data matrix. PC1 explained 55.27% and PC2 35.16% of the data variability. PC1 was mainly characterized by the positive loading of the variables Cd (0.99), Pb (0.99), PAHs (0.79), salinity (0.99), pH (0.83), and DO (0.99) and by the negative loading of the variables NRRT (−0.79) and Cr (−0.99).

To examine the associations between NRRT and bioaccumulated contaminants in *M. galloprovincialis* tissue, the parameters ∑PAH, ∑OCP, ∑PCB, Cu, Cd, Pb, Ni, Cr, Co, and NRRT were included in the multivariate analyses. The PCA showed that PC1 and PC2 explained 98.24% of the total variance in the data matrix (Figure 7). PC1 explained 76.93% and PC2 21.31% of the data variability. The eigenvalues for the first two principal components were 7.69 (PC1) and 2.13 (PC2). PC1 was mainly characterized by the positive loading of the variables Cu (0.97), Pb (0.97), Ni (0.97), Co (0.97), ∑PAH (0.97), and NRRT (0.89), and by the negative loading of the variables ∑OCP (−0.77) and ∑PCB (−0.77). PC2 was represented mainly by positive loadings, especially the loadings of the variables Cd (0.77), ∑OCP (0.63), and ∑PCB (0.63), and by the negative loading of the variable Cr (−0.63).

## 4. Discussion

LMS was selected as a biomarker of cellular distress due to its extensive use as an indicator of contaminant exposure and effect and rapid response at low levels of chemical concentration. The hemocytes (blood cells) of the marine mussel *M. galloprovincialis*, a commercially and ecologically important organism, were used as an experimental tool in this study. Water quality and the degree of contaminant bioaccumulation in the soft tissue of mussels in the study area were analyzed as supporting parameters.

### 4.1. Contaminant Concentrations in Seawater

Seawater concentrations of pollutants provide insight into contaminant input and water quality. The pollutant concentration in seawater showed an increased spatial variability due to the site-specific natural and anthropogenic pressures.

Total petroleum hydrocarbons (TPHs), which are common contaminants in the environment, include several hundred hydrocarbon compounds that originate from the distillates of crude oil in the form of kerosene, gasoline, diesel, motor oil, solvents, etc. [29]. Higher levels of TPHs in seawater were measured at the ports due to intense naval traffic, long-term leakage, accidental spills from ships, industrial and urban sewage discharges, or operational failures in naval activities. Consequently, the TPH level at MDP exceeded 6.2 times the maximum admissible value (200 µg/L) stipulated by national legislation [28]. The selected sampling site was in Berth 4 of the port, mainly used for the supply of crude oil for the nearby oil refinery. This area is particularly prone to accidental spills. A recent example is the accidental spill of approximately one tone of fuel oil (a residue from crude-oil distillation), produced in October 2021, during the fuel unloading activities of a Maltese ship in this berth [30].

According to Oros et al. [31], a ∑PAH below 0.60 µg/L indicates moderate pollution. In our study, the highest ∑PAH at CTP suggested a high pollution level, probably due to accidental ship spills, operational spillage, ship discharges, etc. The phenanthrene and anthracene compounds in seawater exceeded the Maximum Acceptable Concentration (MAC) regulated by national legislation [28], suggesting petroleum exposure. The ∑PAH concentration recorded at the other sampling sites indicated low pollution levels for this class of contaminant.

According to a previous study, the OCP and PCB concentrations in marine waters have shown a decreasing tendency in the last few years [31]. Instead, very high concentrations of OCPs and PCBs were measured in coastal waters in our study, mainly at MGP and MPB. Lindane; heptachlor; cyclodiene pesticides (aldrin, dieldrin, endrin); and DDT (*p*,*p*′DDT, *p*,*p*′DDE, *p*,*p*′DDD) concentrations exceeded the MAC according to European legislation [32] at all sampling sites. An explanation for these results could be that the sampling locations were in the proximity of pollution sources such as urban sewage treatment plants, potentially polluted freshwater inputs, and intense marine traffic.

The HM concentrations were below the Environmental Quality Standards (EQS) for marine waters [32] at all sampling sites. Similar findings are reported in the literature [31,33]. The highest HM concentration in seawater was measured at MPB, a beach area influenced by urban wastewater discharges, polluted freshwater input from a nearby lake, and fishing boats.

The result of the PCA suggested that the LMS response induced in mussels from CTP, MGP, and MDP was caused by the presence of different contaminants at higher concentrations in the seawater. The PCA showed that the NRRT was negatively correlated with Ni and PAHs and positively correlated with Cr, Cu, and OCP concentrations. Regarding the location of sites, PC1 showed a separation between MPB and the other three sites (CTP, MGP, and MDP), indicating a distinct LMS response in these sites.

### 4.2. Tissue Concentrations and Accumulation of Contaminants

The tissue concentrations of pollutants showed large variations across sampling sites. Low tissue levels of PAHs were measured at all sampling sites, in contrast with the results of previous studies carried out along the Black Sea coast [34,35] and in other seas [36,37].

Increased OCP and PCB values were measured at all sampling sites, particularly in port areas, indicating a high level of pollution in the shore waters. Our results were very high compared with other findings recorded in the deeper waters of the Black Sea [34,35]. Other studies carried out in different seas have reported lower tissue levels of OCPs and PCBs compared with our results [36,37,38]. Heptachlor exceeded the MAC according to European legislation [32] at all sampling sites. Generally, PCB 52, PCB 101, PCB 18, and PCB 180 exceeded the Environmental Assessment Concentrations (EACs) established by the OSPAR Commission [39] at all sites.

The tissue levels of Cu, Cd, Pb, Cr, and Ni at MPB and CTP were among the highest recorded in this study. However, the Cd and Pb concentrations were below the MAC according to EU regulations [40]. Our study indicated the low-to-moderate HM contamination of mussels at all sampling sites. The results agreed with previous studies carried out along the Romanian Black Sea coast [31,35,41]. The HM levels found in our study were similar to those from the Turkish Black Sea coast [35] and Baltic Sea [9], higher than those from the southeastern Adriatic Sea [1], and relatively low compared with the results of other studies carried out in the Gulf of Biscay [37] and Mediterranean Sea [36].

The bioaccumulation of organic pollutants and HMs in mussel tissue depends on the environmental concentrations but also a range of biological (i.e., body size, age, and the physiological and biochemical status of the organism) and environmental factors [42,43,44]. Environmental parameters, in particular temperature, salinity, and pH, influence the concentration, distribution, and bioavailability of chemicals in the water column, sediment, and biota [42]. Since this study was carried out in summer, elevated seawater temperatures were recorded at all sampling sites, and this probably influenced the mussels’ contaminant uptake. Low salinity increases toxic free metal ion bioavailability and the toxicity of HMs [45].

Field and experiment studies on mussels have shown that contaminant bioaccumulation in mussels and body size are negatively correlated [46,47,48,49]. Size-dependent contaminant bioaccumulation is attributed to the relationship between metabolic rates and contaminant uptake or loss rates. Smaller mussels compensate for higher metabolic demands by increasing their pumping, filtering, and respiration rates, thereby increasing their exposure to water-borne contaminants [46,48]. The mussels collected from MPS had a small body size (4–5 cm shell length) compared with the mussel specimens sampled from the port areas, which reached 8–9 cm in length. The high bioaccumulation of HMs and PAHs in mussels at MPB could have been related to the low environmental salinity, small body size, increased metabolic demand, and low food availability. In the last decade, a phytoplankton decline has been observed in the shallow waters of Mamaia Bay, the annual average density of the phytoplankton community staying below 1 × 10^6^ cells/L [50]. Our results agreed with the findings of Hervé-Fernandez et al. [50] showing that low salinity and reduced food availability increase the assimilation of Cd by mussels. Turja et al. [9] also suggested that Cd and Cr tissue concentrations could be linked to lower salinity.

The main hydrocarbons accumulated in mussels are the LMW PAHs, the most bioavailable and lipophilic 3-to-5-ringed compounds [51]. Phenanthrene and anthracene, compounds with 2–3 aromatic rings, were the most bioaccumulated compounds at all sites. These findings are supported by various experimental and field studies [9,13,35].

The elevated bioaccumulation of OCPs and PCBs, highly toxic and lipophilic OCs, could be linked to the high lipid content of pre-spawning mussels, high filtration rates associated with increased food availability (in ports), and body size (at MPB) [49]. The ports selected for the present study were permanently enriched by the addition of nutrients due to the discharges of urban and industrial effluents [51] These conditions and the stagnation of the port water favor the development of phytoplankton and can even cause phytoplankton blooms in certain climatic conditions (e.g., during periods of calm weather and greater sunshine). The phytoplankton blooms are more frequent in ports, where the density can reach up to 3.72 × 10^6^ cells/L (e.g., at MGP) [51].

The PCA results revealed that the LMS of mussels from MDP and MGP was more affected by the presence of OCPs and PCBs than those from MPB and CTP. The LMS of mussels from MPB was mainly influenced by the presence of HMs (Cu, Ni, Pb, Co, and Cr) and PAHs. The PCA showed that the NRRT was negatively correlated with OCPs and PCBs and positively correlated with HM and PAH concentrations.

### 4.3. Lysosomal Response to Contaminant Levels

LMS has been acknowledged as one of the most sensitive biomarkers of cellular stress for xenobiotic contamination and has been applied widely in field and experimental studies [2,3,5,7,8,10,13]. Lysosomes are established targets for metal ions and organic pollutants that are also known to accumulate in lysosomes and affect the lysosomal structure [18]. Lysosomes exhibit low membrane stability when overloading their storage capacity [11,12,18,52]. The phagocytic activity of hemocytes is stimulated by some pollutants at low concentrations, whereas prolonged exposure and/or higher chemical levels lead to a reduced ability to take up particles [53]. In the presence of complex mixtures of contaminants, the phagocytic activity of hemocytes is affected by both stimulating and inhibitory mechanisms [54]. Exposure to various contaminants, both metals and organic pollutants, can increase cellular radical generation [55,56,57,58]. Lysosomal integrity and function are adversely affected due to exposure to various environmental pollutants [14]. A significantly decreased NRRT has been confirmed in response to metal exposure [56], PAHs [14], and high levels of PCBs [1]. Low-molecular-weight PAHs have a significant acute toxic effect on mussels and cause lysosomal destabilization, which is believed to be directly related to the mechanism of cell cytotoxicity [2,13,52,59].

According to the assessment criteria established for the neutral red retention test [5], none of the mussel groups in this study indicated a “healthy” status (NRRT ≥ 120 min). Mussels from all sites could be assessed as “severely stressed” (NRRT < 50 min) and exhibiting pathologies (i.e., lysosome enlargement, lysosomes leakage, rounded up fragmenting cells) as described by Viarengo et al. [6]. The low LMS observed at all sampling sites could be mainly linked with the higher levels of OCPs and PCBs. Exposure to a diverse mixture of chemicals in the environment enhances toxic effects [2]. In our study, the effects on LMS were seen even at low concentrations of some pollutants, suggesting that the complexity of contaminant mixtures had a greater toxic effect regardless of the individual pollutant concentrations. We should also take into consideration the potential effects on the lysosomal response of the poorly or non-investigated chemical compounds present in the seawater (i.e., pharmaceuticals, personal care products, detergents, biocides, and aliphatic hydrocarbon compounds).

The low lysosomal stability observed in this research was similar to that observed in other studies conducted along the Romanian Black Sea coast on *M. galloprovincialis* [19,20]. Ciocan [19] showed an exceptionally low NRRT (0 min) in the hemocytes of mussels collected from Navodari and Mamaia Pescarie Bay. Moore et al. (1999) demonstrated that mussels collected from the Romanian coast (in Navodari and Mamaia Pescarie Bay) had the lowest NRRT (5.4–12 min) compared to other regions in the Black Sea (120–150 min). Another NRRT assay performed on clam hemocytes (*Mya arenaria)* also showed low NRRTs (0–15 min) in a polluted site [21]. The results obtained in the present study were particularly low compared to the data obtained in other field studies [3,5,7].

The reduced LMS obtained in this research indicated stress and damage in the hemocytes of mussels following environmental contaminant exposure. LMS is an effective indicator of health status in mussels of the Romania Black Sea coast, and it could be related to tissue concentrations of pollutants. Lysosomal damage, evaluated in *M. galloprovincialis* hemocytes by the neutral red retention assay, is a sensitive biomarker of cellular stress induced by contaminant exposure. It may also serve as an early warning for contaminant-induced stress. This study provided an extended ecotoxicological evaluation and baseline data of the contaminant-induced response in mussels from the Romania Black Sea coast. Furthermore, this study confirmed the need for a suite of biomarkers to test ecotoxicological responses to contaminant exposure in the marine environment.

## 5. Conclusions

This research demonstrated the utility of lysosomal membrane stability (LMS) as a biomarker for cellular distress in marine mussels exposed to contaminants. The study focused on the hemocytes of the marine mussel *M. galloprovincialis*, a commercially and ecologically important organism. Water quality and the accumulation of contaminants in the mussel tissue were analyzed to support the results.

LMS was used as a sensitive biomarker to assess cellular stress induced by contaminants exposure. The results showed that mussels from all sampling sites exhibited severely stressed conditions, indicating the potential presence of pathologies. The low LMS observed in the mussels was primarily associated with higher levels of OCPs and PCBs, suggesting the toxic effects of contaminant mixtures.

The complexity of contaminant mixtures in the environment may have a greater impact on lysosomal stability than individual pollutant concentrations. Future research should consider the potential effects of poorly or non-investigated chemical compounds present in seawater on the lysosomal response.

Overall, this research provides important insights into the contamination levels and cellular responses of marine mussels on the Romanian Black Sea coast. The findings highlight the need for a comprehensive suite of biomarkers to assess the ecotoxicological impacts of contaminant exposure in the marine environment. Future research should investigate the mechanisms underlying lysosomal damage and explore additional biomarkers to enhance our understanding of the health status of marine organisms exposed to contaminants. Furthermore, the continuous monitoring and assessment of contaminant levels in the marine environment are essential for effective environmental management and conservation efforts.

## Figures and Tables

**Figure 1 toxics-11-00649-f001:**
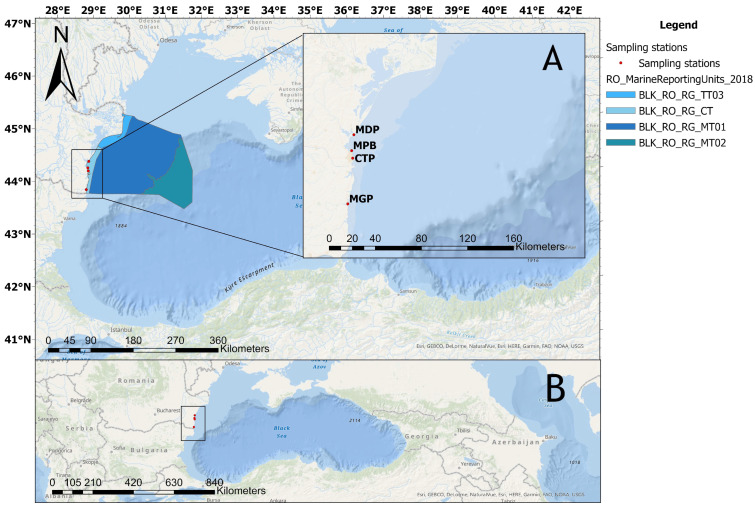
Map of the study area. The square shows the location of the sampling sites on the Romanian coast (**A**) and the position in the Black Sea region (**B**). MDP: Midia Port; MPB: Mamaia Pescarie Bay; CTP: Constanta Port; MGP: Mangalia Port.

**Figure 2 toxics-11-00649-f002:**
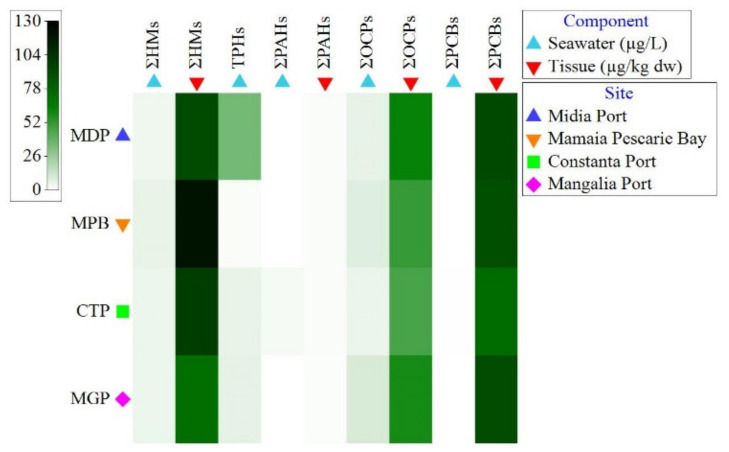
Contaminant concentrations in seawater and tissues at the sampling sites. Data are expressed as square-root-transformed mean values. ∑PAH: sum of polycyclic aromatic hydrocarbons; ∑OCP: sum of organochlorine pesticides; ∑PCB: sum of polychlorinated biphenyls. The intensity of the green scales is proportional to the contaminant concentration. White or lightly colored areas indicate low concentrations of pollutants at that site.

**Figure 3 toxics-11-00649-f003:**
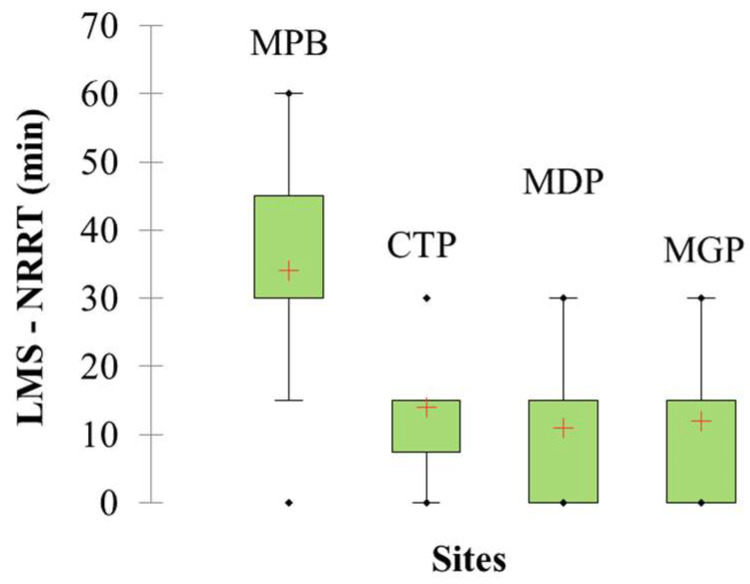
Box–whisker plot of lysosomal membrane stability (LMS) in hemocytes of mussels assessed by the neutral red retention time (NRRT) (*n* = 15). MDP: Midia Port; MPB: Mamaia Pescarie Bay; CTP: Constanta Port; MGP: Mangalia Port. Box plot legend: plus symbol—mean value; box—data range of 25–75%; points—outliers; whiskers—minimum and maximum values.

**Figure 4 toxics-11-00649-f004:**
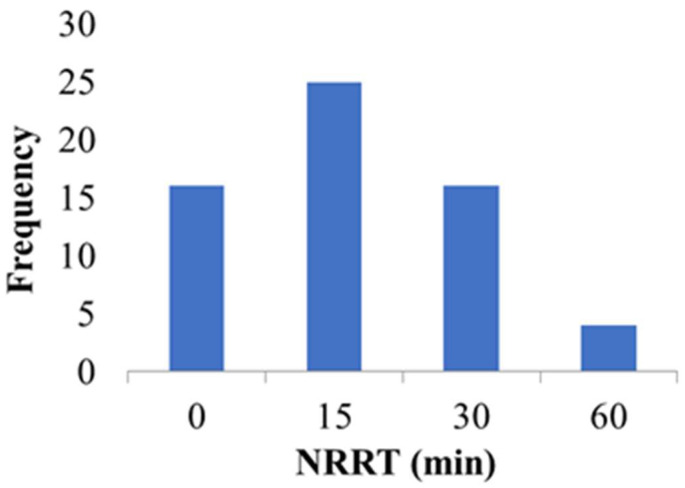
Frequency of neutral red retention time (NRRT) values in mussel specimens (*n* = 60) during the assay.

**Figure 5 toxics-11-00649-f005:**
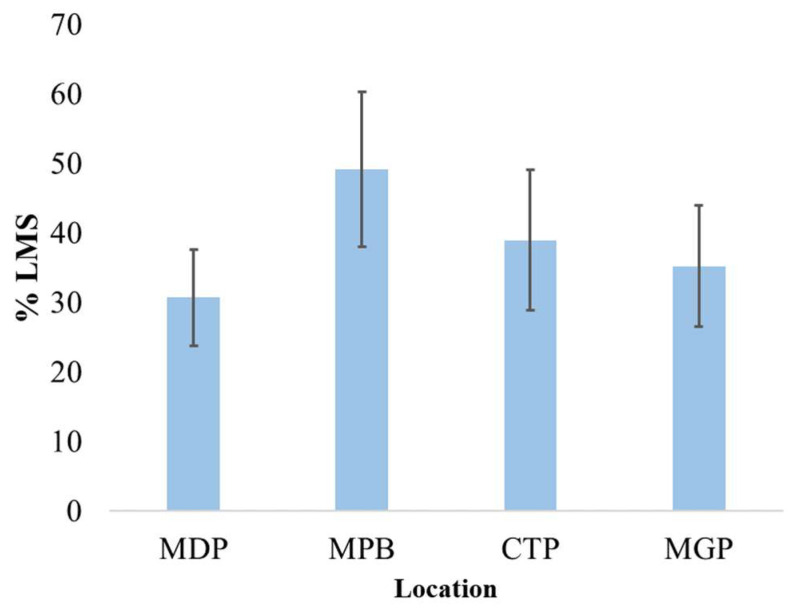
Mean percentage of lysosomal damage (% LMS ± std. dev.) in hemocytes of mussels (*n* = 60). LMS: lysosomal membrane stability. MDP: Midia Port; MPB: Mamaia Pescarie Bay; CTP: Constanta Port; MGP: Mangalia Port.

**Figure 6 toxics-11-00649-f006:**
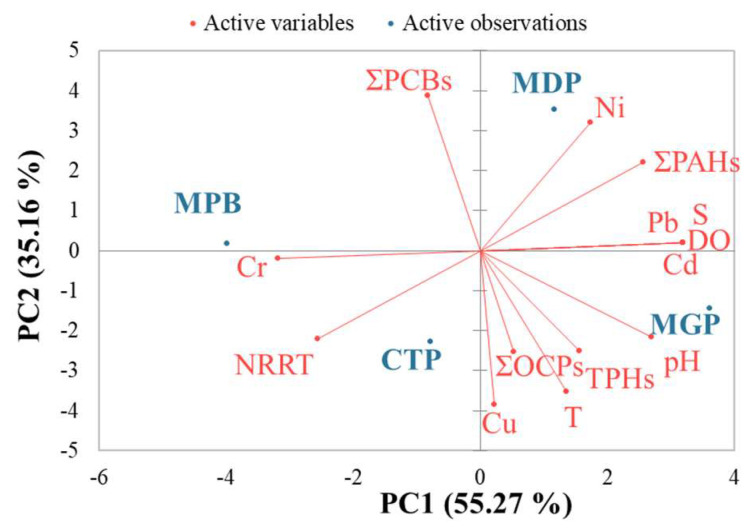
Principal component analysis (PCA) of the contaminants analyzed in seawater, abiotic factors, and by neutral red retention time (NRRT). T: temperature; S: salinity; DO: dissolved oxygen; Cu: copper; Cd: cadmium; Pb: lead; Ni: nickel; Cr: chromium; ∑PAH: sum of polycyclic aromatic hydrocarbons; ∑OCP: sum of organochlorine pesticides; ∑PCB: sum of polychlorinated biphenyls. MDP: Midia Port; MPB: Mamaia Pescarie Bay; CTP: Constanta Port; MGP: Mangalia Port.

**Figure 7 toxics-11-00649-f007:**
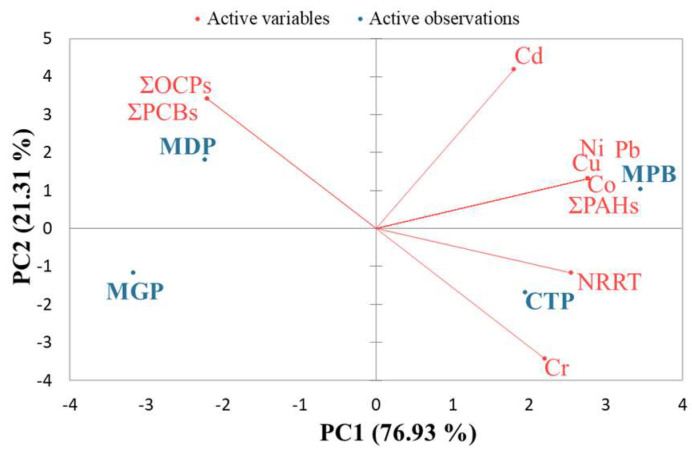
Principal component analysis (PCA) of the contaminants bioaccumulated in mussel tissue and by neutral red retention time (NRRT). Cu: copper; Cd: cadmium; Pb: lead; Ni: nickel; Cr: chromium; Co: cobalt; ∑PAH: sum of polycyclic aromatic hydrocarbons; ∑OCP: sum of organochlorine pesticides; ∑PCB: sum of polychlorinated biphenyls. MDP: Midia Port; MPB: Mamaia Pescarie Bay; CTP: Constanta Port; MGP: Mangalia Port.

**Table 1 toxics-11-00649-t001:** Location of the sampling sites and description of anthropogenic pressures. Lat: latitude; Long: longitude.

Location	Sampling Site Code	Lat. (°N)	Long. (°E)	Anthropogenic Pressures
Midia Port	MDP	44.342436	28.681631	Urban and industrial sewage discharge, oil refinery, petrochemical activities, and intense maritime traffic.
Mamaia Pescarie Bay	MPB	44.219671	28.649602	Urban sewage discharge, moderate maritime traffic, small fishing boat dock.
Constanta Port	CTP	44.160792	28.657107	Urban and industrial sewage discharge, intense maritime traffic, intense cargo operations (crude oil and derivative products, coal, ore, chemicals, fertilizers, etc.)
Mangalia Port	MGP	43.807017	28.582175	Urban sewage discharge, yacht and small boat docks, tourism in summer.

**Table 2 toxics-11-00649-t002:** Physico-chemical parameters of seawater at the sampling sites. MDP: Midia Port; MPB: Mamaia Pescarie Bay; CTP: Constanta Port; MGP: Mangalia Port; T: temperature; S: salinity; DO: dissolved oxygen.

Site	T (°C)	S (psu)	DO (mg/L)	pH
MDP	23.0	14.63	13.56	8.26
MPB	23.6	13.81	5.97	8.07
CTP	23.9	14.09	6.23	8.37
MGP	26.0	15.08	17.80	8.63

## Data Availability

The data belong to the National Institute for Marine Research and Development “Grigore Antipa” (NIMRD) and can be accessed by request to http://www.nodc.ro/data_policy_nimrd.php, accessed on 7 March 2023.

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
