# Peer review of "Lysosomal Membrane Stability of Mussel (Mytilus galloprovincialis Lamarck, 1819) as a Biomarker of Cellular Stress for Environmental Contamination"

_toxics, 2023, doi:10.3390/toxics11080649_

Round 1
Reviewer 1 Report
Review
The manuscript (toxics-2507731):”Lysosomal Membrane Stability of Mussel (Mytilus galloprovincialis Lamarck, 1819) as a Biomarker of Cellular Stress for Environmental Contamination” deals with the impact of pollution on marine mussels based on biological indicators. This comprehensive approach contains data on broad range of organic and inorganic pollutants in the seawater and mussel’s soft tissues. Additionally, the effects of pollutants on mussels were evaluated by neutral red retention time assay to find out the level of cellular stress.
The concept, methodology and the main results are well explained. However, the main drawback of the manuscript is the lack of conclusion. The discussion section is complex and long, so I suggest the authors to make the summary, point out the main conclusions in separate section and to give the reasons why this kind of research should be continued in the Black Sea region.
The authors have to include standard deviations in Figures 4 and 5.
Specific remarks.
Lines 7-18: there is no need to write the name of the same institution 5 times. One time is enough.
Line 80: “total contaminant”: Which kind of contaminants? The authors should add the type of contaminants that they plan to investigate.
Line 85: What kinds of samples were taken? At the section onset, the authors need to give an overview of different samples.
Lines 226-227: The authors should modify this sentence. It is not clear enough.
Line 409: “indicate”. The font size should be modified.
Line 455: It is not common in scientific literature to use “because”. It is better to replace with “since” in this case.
Lines 480-482: LMS assay is not a biomarker. Biomarker is only LMS while LMS assay is the method. Please make correction.
Line 515: “findings” should be replaced with “the results”.
Line 516: “the results” should be replaced with “the data”
The English is fine.
Author Response
Dear Reviewer,
Thank you for providing valuable comments on our research. We have carefully addressed your suggestions and made the necessary changes as outlined in the table. Additionally, we have included a comprehensive conclusion section summarizing our findings. Moreover, we have incorporated Figures 4 and 5, along with the corresponding standard deviation information.
Please see the attachment.
Thank you once again for your valuable feedback, which has significantly improved the quality of our research. We believe that these revisions have strengthened our study and provided a more robust analysis of our results.
Sincerely,
The Authors

Reviewer 2 Report
Line 85: "Four field samplings were conducted along the Romanian Black Sea coast". According to the description of the locations, all seem to suffer anthropogenic pressure or impact. I miss a pristine or little impacted place in order to use it as a kind of control location to compare the obtained data with the rest. I think this is the main flaw of the study. It may happen (I have never been to Romania) that there is no control location but then, you should mention it.
Line 100: "and transported to the laboratory in a cool box with ice packs". How long did it take?
Line 324: "Most specimens (25) retained the NR". 25 specimens out of 35 are not most. Related with this chart (Figure 4), I miss a comment on the group 30 min.
Line 350: Table 3. No necessary to put the table and I suggest to put in supplementary material instead. I also suggest to avoid the statistical data on the text. You have already mention in the data analysis section which the significant level was. So, in the results section I suggest to write if there were significant differences or not and as I mentioned before, send the Table 3 to the supplementary material.
Figure 6 and 7: You do a Principal Component Analysis of the contaminants analysed in seawater and bioaccumulated in mussels but I do not see that the results obtained from them are discussed on the Discussion section.
Line 409: the word "indicate" is not written in a proper size.
Line 468: you mention the "low food availability" in MPB and then in line 478 "increased food availavility (in ports)". Can you add any data or reference about the dynamics of phytoplancton in these areas in order to strengthen this comments?
Line 485: "The phagocytic activity of lysosomes". The lysosomes do not have phagocytic activity. Haemocytes do have phagocytic activity. Phagosomes are vesicles formed around a particle engulfed by phagocytes, in this case hemocytes, via phagocytosis and then phagosomes fuse with lysosomes, resulting in phagolysosomes.
Line 488: "the haemocyte function is affected by both ...". Which function(s)? Please, explain it better.
Line 498-499: "and exhibiting pathologies as described by Viarengo et al. ? Please, mention the pathologies. Lysosomal enlargement, due to a decrease in the stability of the lysosomal mebrane and fusion of lysosomes?
Author Response
Dear Reviewer,
We would like to express our gratitude for your valuable comments. We have taken them into careful consideration and made all the necessary changes to enhance both the research design and the discussion sections.
Please see the attachment
Thank you for your insightful feedback, which we believe has greatly improved the overall quality of our work.
Sincerely,
The Authors

Round 2
Reviewer 2 Report
Dear Authors,
Please find below 3 suggestions forn line 530, 547 and 551.
Line 530: “In the last decade,……of Mamaia Bay, the annual average density of the phytoplancton community staying below 1·106 cells/L”.
Line 547: “The phytoplankton blooms are more frequent in ports where the density can reach up tp 3.72·106 cells/L (e.g., in MGP)[51].”
Line 551: ”….the presence of HMs (Cu, Ni, Pb, Co and Cr) and PAHs. The PCA showed that the NRRT was negatively correlated with OCPs and PCBs, and positively correlated with HMs and PAHs concentrations.
If I mentioned to add standard deviation in Figures 4 and 5, I beg your pardon for the confusion since I meant only Figure 5, which represents a mean percentage and make sense to add standard deviation bars but not in Figure 4, that shows frecuency which can be only represented by discrete numbers and makes no sense at all adding standard deviation bars.
Finally, I suggest to decrease the size of the Conclusions beeing more precise in your comments. Remember what the other reviewer said about LMS assay, this is not a biomaker but the method employed to detect if the stability of the lysosomal membrane (biomarker) increases or decreases.
They are suggested in the comments.
Author Response
Dear Reviewer,
We sincerely appreciate your valuable comments and have made the necessary changes as per your suggestions.
Thank you for your time and consideration.
Sincerely,
The Authors
